# Genome-Based Reclassification of [*Bizionia*] *algoritergicola* Bowman and Nichols 2005 as *Algorimicrobium algoritergicola* gen. nov., comb. nov. and description of *Algorimicrobium bowmanii* sp. nov.

**DOI:** 10.3390/microorganisms14010024

**Published:** 2025-12-21

**Authors:** Valeriya Kurilenko, Evgeniya Bystritskaya, Nadezhda Otstavnykh, Peter Velansky, Sergey Baldaev, Viacheslav Eremeev, Natalya Ageenko, Konstantin Kiselev, Olga Nedashkovskaya, Marina Isaeva

**Affiliations:** 1G.B. Elyakov Pacific Institute of Bioorganic Chemistry, Far Eastern Branch, Russian Academy of Sciences, Prospect 100 Let Vladivostoku, 159, Vladivostok 690022, Russia; ep.bystritskaya@yandex.ru (E.B.); chernysheva.nadezhda@gmail.com (N.O.); baldaevsergey@gmail.com (S.B.); wieremeew@gmail.com (V.E.); oned2004@mail.ru (O.N.); 2A.V. Zhirmunsky National Scientific Center of Marine Biology, Far Eastern Branch, Russian Academy of Sciences, Palchevskogo Street 17, Vladivostok 690041, Russia; velansky.pv@gmail.com (P.V.); natkuprina@mail.ru (N.A.); 3Federal Scientific Center of the East Asia Terrestrial Biodiversity, Far Eastern Branch of the Russian Academy of Sciences, Vladivostok 690022, Russia; kiselev@biosoil.ru

**Keywords:** sea urchin, *Algorimicrobium algoritergicola* gen. nov., comb. nov., *Algorimicrobium bowmanii* sp. nov., genomic analysis, Sea of Okhotsk

## Abstract

The genus *Bizionia*, a member of the *Bacteroidota* phylum, is considered a polyphyletic taxonomic group requiring a phylogenetic revision of its members. A novel strain 041-53-Ur-6^T^ was isolated from the cavity fluid of the sea urchin *Strongylocentrotus intermedius* from the Sea of Okhotsk. Analysis of the 16S rRNA gene sequence showed that 041-53-Ur-6^T^ belongs to the family *Flavobacteriaceae*, and its closest neighbor is [*Bizionia*] *algoritergicola* with 97.5% sequence similarity. Phylogenomic analysis confirmed the phylogenetic heterogeneity of the genus *Bizionia* and the clear separation of the genera “*Algorimicrobium*” and *Hanstruepera*. The inter-genus AAI values between them were 74.0–76.4%, which is slightly lower than the inter-species AAI values observed for each genus. The strain 041-53-Ur-6^T^ (= KMM 8389^T^) formed a separated branch within the [*B.*] *algoritergicola* clade, demonstrating the highest ANI/AAI values of 80.1/81.0% with the strain [*B.*] *algoritergicola* APA-1^T^. The dDDH values between strain KMM 8389^T^ and representatives of the genus “*Algorimicrobium*” ranged from 22.6% to 26.7%. Major fatty acids were iso-C_15:1_ *ω10c*, iso-C_15:0_ and iso-C_15:0_ Δ2-OH. The polar lipids included a phosphatidylethanolamine, a phosphatidylglycerol, five unidentified lipids, two unidentified aminolipids, a phosphatidylcholine, and an unidentified aminophospholipid. The genome KMM 8389^T^ is a circular chromosome of 3,031,910 bp in size with a DNA G + C content of 33.5%. It comprises 2702 protein-coding genes and four *rrn* operons. Functional genomic analysis indicated the potential of KMM 8389^T^ for degrading starch, glycogen, and alginate due to the presence of genes encoding GH13, GH31, and GH65. Furthermore, KMM 8389^T^ possessed PLs 6, 7, 12, and 17, specialized for alginate, confirming the potential adaptation of this strain to algal substrates and surfaces. On the basis of the results of genotypic, chemotaxonomic, and phenotypic analyses, it is clear that the strain KMM 8389^T^ represents a novel species with [*B*.] *algoritergicola*, [*B.*] *argentinensis*, [*B.*] *echini*, [*B.*] *hallyeonensis*, [*B.*] *myxarmorum*, [*B.*] *psychrotolerans*, and [*B.*] *sediminis* as the nearest neighbors. These taxa are classified in a single novel genus, as *Algorimicrobium algoritergicola* gen. nov., comb. nov., *A*. *argentinensis* comb. nov., *A*. *echini* comb. nov., *A*. *hallyeonensis* comb. nov., *A*. *myxarmorum* comb. nov., *A*. *psychrotolerans* comb. nov., *A*. *sediminis* comb. nov., and *Algorimicrobium bowmanii* sp. nov. 041-53-Ur-6^T^ (=KMM 8389^T^, =KCTC 72011^T^).

## 1. Introduction

The *Bacteroidota* phylum is a cosmopolitan phylum with members found throughout a variety of habitats on Earth. A distinctive feature of these bacteria is the ability to decompose carbohydrates. Classification of the phylum *Bacteroidota* is a challenge requiring the use of comparative genomic analysis. Marina Gorcia-Lopez and co-authors performed a phylogenetic analysis of 1000 genomes of the phylum *Bacteroidota* (formally “*Bacteroidetes*”) [1]. The authors showed that some taxa require revision, for example, the genus *Bizionia,* which is non-monophyletic. In the 16S rRNA gene tree, the two species, [*B.*] *echini* and [*B.*] *argentinensis*, formed a well-supported clade together with [*B.*] *algoritergicola*, [*B.*] *hallyonensis*, [*B.*] *myxarmorum*, [*B.*] *psychrotolerans*, and *B. sediminis*. While this clade was generally well supported, support for the monophyly of the entire genus *Bizionia* was no longer present in the published 16S rRNA gene trees used to describe the most recent species. Obtaining monophyletic genera by merging all the hybrid genera with *Bizionia* would introduce a huge number of changes and is also considered inappropriate, given the genomic and phenotypic divergence of the group. Thus, it was proposed to combine the strains of [*B.*] *algoritergicola*, [*B.*] *argentinensis*, [*B.*] *echini*, [*B.*] *hallyeonensis,* [*B.*] *myxarmorum*, [*B.*] *psychrotolerans*, and [*B.*] *sediminis* into a new genus named “*Algorimicrobium*” in honor of [*B.*] *algoritergicola*, as the first species described from this group.

The genus *Bizionia* was established by Nedashkovskaya et al. (2005) [2]. To date, 13 species with valid nomenclature have been described in the genus *Bizionia* (https://lpsn.dsmz.de/genus/bizionia, accessed on 26 November 2025). These bacteria have been isolated from Antarctic surface seawater, coastal Antarctic areas, and various marine invertebrates (a mussel, a coral, a sea cucumber, and a sea urchin), as well as from fish [2,3,4,5].

In this work, we report the isolation and identification of novel halotolerant, Gram-negative, aerobic, and orange-pigmented marine bacteria. On the basis of the results of genotypic, chemotaxonomic, and phenotypic analyses, it is clear that the isolates represent a novel species, with [*B*.] *algoritergicola* CIP 108533^T^ [6], [*B.*] *argentinensis* JUB59^T^ [7], [*B.*] *echini* KMM 6177^T^ [8], [*B.*] *hallyeonensis* KCTC 23881^T^ [9], [*B.*] *myxarmorum* CIP 108535^T^ [6], [*B.*] *psychrotolerans* KCCM 43042^T^ [10], and [*B.*] *sediminis* KCTC 42587^T^ [11] as the nearest neighbors. These taxa are classified in a single novel genus, such as *Algorimicrobium algoritergicola* gen. nov., comb. nov., *A*. *argentinensis* comb. nov., *A*. *echini* comb. nov., *A*. *hallyeonensis* comb. nov., *A*. *myxarmorum* comb. nov., *A*. *psychrotolerans* comb. nov., *A*. *sediminis* comb. nov., and *Algorimicrobium bowmanii* sp. nov.

## 2. Materials and Methods

### 2.1. Isolation and Phenotypic Characterization of Bacteria

The sea urchin *Strongylocentrotus intermedius* was collected by bottom trawl at a depth of 400 m in July 2011 during scientific expedition No. 41 on the R/V “Akademik Oparin” in the Vries Strait (45.500000, 148.941668), Iturup Island, Kuril Islands, Sea of Okhotsk, Russia. Strain 041-53-Ur-6^T^ was isolated from the cavity fluid of this sea urchin and stored at −70 °C in the Difco^TM^ Marine Broth 2216 (Becton Dickenson, Franklin Lakes, NJ, USA) (MB 2216) supplemented with 20% (*v*/*v*) glycerol. Strain 041-53-Ur-6^T^ was deposited to the Collection of Marine Microorganisms (KMM), G. B. Elyakov Pacific Institute of Bioorganic Chemistry (PIBOC), Far Eastern Branch of the Russian Academy of Sciences (FEB RAS), Russia, under the number of KMM 8389^T^ and to the Korean Collection for Type Cultures (KCTC), Korea Research Institute of Bioscience and Biotechnology, Republic of Korea, under the number of KCTC 72011^T^. The type strains, [*B*.] *algoritergicola* CIP 108533^T^ (= APA-1^T^ = ACAM 1056^T^), [*B.*] *myxarmorum* CIP 108535^T^ (= ADA-4^T^ = ACAM 1058^T^), and [*B.*] *echini* KMM 6177^T^ (= KCTC 22015^T^ = LMG 25220^T^) were kindly provided by Nedashkovskaya O. I. The type strains, [*B*.] *algoritergicola* CIP 108533^T^ (= APA-1^T^ = ACAM 1056^T^) and [*B.*] *myxarmorum* CIP 108535^T^ (= ADA-4^T^ = ACAM 1058^T^), were deposited at the KMM, PIBOC FEB RAS, under numbers KMM 8430^T^ and KMM 8431^T^, respectively. All strains used in this study for phenotypic tests and lipid analyses were grown on/in MB 2216 and MA 2216, if not stated otherwise. Gram-staining was examined according to the standard method, oxidase activity was determined using tetramethyl-p-phenylenediamine, and catalase activity was determined using 3% hydrogen peroxide [12]. Gliding motility was observed according to the method of Bowman [13]. The morphology of cells negatively stained with a 1% phosphotungstic acid was examined using electronic transmission microscopy Libra 120 (Carl Zeiss, Oberkochen, Germany), provided by A. V. Zhirmunsky National Scientific Center of Marine Biology, FEB RAS, using cells grown in MB 2216 on carbon-coated 200-mesh copper grids. The tests, including hydrolysis of starch, gelatin, L-tyrosine, chitin, casein, DNA, and nitrate reduction (sulfanilic acid/α-naphthylamine test), growth at different salinities (0–10% NaCl), temperatures (5–40 °C), and pH values (4.0–11.5) were carried out as described by Smibert and Krieg (1994) [12]. The medium MA 2216 (or MB 2216) was used as a basal, while mannitol and CaCO_3_ were omitted for the determination of substrate hydrolysis and pH, respectively. Formation of H_2_S from thiosulfate was tested in the MB 2216 using a lead acetate paper strip. Biochemical tests for all studied strains and KMM 8389^T^ using API 20E, API 20NE, and API ZYM (bioMérieux, Marcy-l’Étoile, France) were performed as described by the manufacturer [14]. Carbon source utilization was performed with the API 50 CHB/E tests (bioMérieux, Marcy-l’Étoile, France) according to the manufacturer’s instructions [15].

Antibiotic susceptibility of strains studied was examined on MA 2216 plates using commercial paper discs (Research Centre of Pharmacotherapy, St. Petersburg) impregnated with the following antibiotics (µg per disc, unless otherwise indicated): ampicillin (10), benzylpenicillin (10 U), vancomycin (30), gentamicin (10), kanamycin (30), carbenicillin (100), chloramphenicol (30), neomycin (30), oxacillin (10), oleandomycin (15), lincomycin (15), ofloxacin (5), rifampicin (5), polymyxin (300 U), streptomycin (30), cephazolin (30), cephalexin (30), erythromycin (15), nalidixic acid (30), tetracycline (30), and doxycycline (10). For polar lipid and fatty acid analyses, strain KMM 8389^T^ and one related type strain were cultivated on MA 2216 at 24 °C for 24 h. Lipids were extracted using a chloroform–methanol–water (2:2:1, by vol.); after phase separation, the lower chloroform layer was collected [16]. Two-dimensional thin layer chromatography of polar lipids was carried out on Silica gel 60 F 254 (10 × 10 cm, Merck, Darmstadt, Germany) using chloroform–methanol–water (65:25:4, *v*/*v*) for the first direction, and chloroform–methanol–acetic acid–water (80:12:15:4, *v*/*v*) for the second one [17]. Lipids were detected by sequentially spraying with 0.25% ninhydrin in acetone (for detecting amino group-containing lipids) [18], molybdate reagent (for detecting phospholipids) [19], and 5% sulphuric acid in methanol, followed by heating at 130 °C [18]. Respiratory lipoquinones were analyzed by the reversed-phase HPLC using a modified method [20]. A Shimadzu LC–30 chromatograph with a photodiode array detector (SPD–M30A), equipped with Shimpack ODS II (150 × 2.1 mm) column, was used. The column temperature was 40 °C, isocratic elution with methanol–isopropanol (7:3) with the addition of 0.1% of formic acid was used. Fatty acid methyl esters (FAMEs) were prepared according to the procedure of the Microbial Identification System (MIDI) [21]. The quantitative analysis of FAMEs was performed using the GC–2010 chromatograph (Shimadzu, Kyoto, Japan) equipped with a capillary column SH–Rtx–5 ms (30 m × 0.25 mm I.D.) (Shimadzu, Kyoto, Japan). The temperature was programmed from 160 °C to 250 °C, at a rate of 2 °C/min. Identification of FAMEs was accomplished by equivalent chain length values and by comparing the retention times of the samples to those of standards. In addition, FAMEs were analyzed using a GC-MS Shimadzu model QP2020 (column SH–Rtx–5 ms, the temperature program from 160 °C to 250 °C, at a rate of 2 °C/min).

### 2.2. 16S rRNA Gene Sequence and Phylogenetic Analysis

The DNA of strain 041-53-Ur-6^T^ (= KMM 8389^T^) was extracted using the NucleoSpin Tissue kit (Macherey–Nagel, Düren, Germany) and used for PCR-amplification and sequencing of the 16S rRNA gene as described previously [22]. The obtained sequence was compared with the 16S rRNA gene sequences of validly published type strains using the nucleotide similarity search tool on the EzBioCloud server, accessed on 20 November 2025 [23]. The 16S rRNA phylogenetic relationships of strain KMM 8389ᵀ, along with those of closely related type strains, were estimated by the GGDC web server (http://ggdc.dsmz.de/, updated on 20 November 2025) [24] using the DSMZ phylogenomics pipeline [25]. Maximum likelihood (ML) and maximum parsimony (MP) trees were inferred from the alignment with RAxML [26] and TNT [27], respectively, with bootstrap analysis of 1000 replicates.

### 2.3. Whole-Genome Sequencing and Genome-Based Phylogenetic Analysis

The DNA library for KMM 8389^T^ was prepared with Nextera DNA Flex kit (Illumina, San Diego, CA, USA) and subsequently sequenced on an Illumina MiSeq instrument using paired-end runs with a 150 bp read length. The nanopore library was obtained with SQK-NBD114.96 kit (Oxford Nanopore Technologies, Oxford, UK) and sequenced on the MinION, flow cell FLO-MIN 114 (Oxford Nanopore Technologies, Oxford, UK). Basecalling was performed using Dorado (v. 1.0.2). Resulting short and long reads were trimmed and filtered using Trimmomatic (quality over 30, length over 100) v. 0.39 [28] and chopper (quality over 16, length over 2000, v. 0.10.0) [29], respectively. The quality of processed reads was assessed with FastQC v. 0.11.8 (https://www.bioinformatics.babraham.ac.uk/projects/fastqc/, accessed on 10 June 2025). The filtered reads were used for hybrid assembly with Autocycler v. 0.5.0 [30]. The pipeline facilitated the obtainment of four subsets of long reads, which were then assembled independently using flye v. 2.9.2 [31], canu v. 2.2 [32], miniasm v. 0.3 [33], nextdenovo v. 2.5.1 [34], plassempler v. 1.8.0 [35], and raven v. 1.8.3 [36] as advised by the Autocycler manual. Additionally, the same four subsets underwent a hybrid assembly with a set of short reads using unicycler v. 0.5.0 [37] with default parameters. The resulting assemblies were then combined into a single consensus assembly, which was additionally polished with pilon v. 1.24 [38]. Sequencing depth was estimated utilizing samtools v. 1.3 [39]. The genome completeness and contamination were assessed with CheckM v. 1.1.3 [40]. Gene annotation was performed with RAST [41] and NCBI PGAP [42]. The chromosomal replication origin was located using Ori-Finder 2022 [43].

Genome-based phylogeny was performed using PhyloPhlAn v. 3.0.1 [44] with the provided dataset of 400 conserved proteins, and an ML tree was reconstructed by RAxML v. 8.2.12 [26] under the LG + Γ model with non-parametric bootstrapping of 100 replicates. The average pairwise values of Nucleotide Acid Identity (ANI), Amino Acid Identity (AAI), and in silico DNA–DNA hybridization (dDDH) were estimated by fastANI [45], EzAAI [46], and TYGS platform [47], respectively.

### 2.4. Functional Genomic Analysis

The pan-genome of the “*Algorimicrobium*” and *Bizionia* type strains with metabolism analysis was carried out using the Anvi’o workflow v. 8 as described at https://merenlab.org/2016/11/08/pangenomics-v2/, accessed 21 November 2025 [48].

Carbohydrate-active enzymes (CAZymes) and CAZyme-containing gene clusters were automatically annotated by an online server, dbCAN3 v. 10 (http://bcb.unl.edu/dbCAN2/, accessed on 10 November 2025) [49,50]. Biosynthetic gene clusters of secondary metabolites were identified and annotated using the antiSMASH server, v. 8. (https://antismash.secondarymetabolites.org, accessed on 10 November 2025) [51]. Identification of the Secretion Systems components was conducted with MacSyFinder v. 2.1.4 (TXSScan-1.1.3) [52]. The heat maps and bar plots were visualized using the pheatmap v. 1.0.12 and ggplot2 v. 3.5.1 packages in RStudio v. RStudio/2024.09.1+394 with R v. 4.4.2. Fonts and sizes in all figures were edited manually in Adobe Photoshop CC 2018 for better visualization. The functional and ecological analyses of the strain were performed using the Protologger web tool [53], https://www.protologger.de/, accessed 10 November 2025.

## 3. Results and Discussion

### 3.1. Phylogenetic Analyses

To roughly estimate the taxonomic status of strain 041-53-Ur-6^T^ (= KMM 8389ᵀ), the 16S rDNA sequence was amplified (1368 bp long), sequenced, and compared with those of type strains on the EzBioCloud server [23]. The highest sequence similarity of 97.3–97.5% was revealed with [*B.*] *echini* DSM 23925^T^, [*B.*] *hallyeonensis* T-y7^T^, and [*B.*] *algoritergicola* APA-1^T^. Similarity values with other validly published *Bizionia* species were between 95.2% (*B. paragorgiae* KMM 6029^T^) and 97.0% ([*B.*] *psychrotolerans* PB-M7^T^). It indicated that KMM 8389^T^ can represent a novel species of the genus *Bizionia*.

To determine the phylogenetic position of KMM 8389ᵀ among type strains of the genus *Bizionia* and some closely related genera, 16S rRNA phylogenetic trees were reconstructed (Figure 1). On these trees, KMM 8389ᵀ formed a distinct branch inside the [*B.*] *algoritergicola* clade (“*Algorimicrobium*”). Furthermore, on the *Bizionia* 16S rRNA tree (Appendix A), KMM 8389ᵀ was clearly grouped with the [*B.*] *algoritergicola* clade, as confirmed by high bootstrap values.

It has been known that the monophyly of the genus *Bizionia* is not supported by 16S rRNA data [1]. The observed taxonomic discrepancies concern [*B.*] *algoritergicola*, [*B.*] *argentinensis*, [*B.*] *echini*, [*B.*] *hallyeonensis*, [*B.*] *myxarmorum*, [*B.*] *psychrotolerans*, [*B*.] *sediminis*, and [*B.*] *arctica*; the first eight species are grouped into one clade (the [*B.*] *algoritergicola* clade), which is closer to the type species *Hanstruepera neustonica* [54] than the type species *B. paragorgiae* [2]. The phylogenetic position of [*B.*] *arctica* is uncertain due to its clustering with some type strains of *Formosa* and *Xanthomarina* caused by the low resolution of 16S rRNA genes. While genome sequences were available only for three *Bizionia* species, the phylogenomic analysis showed that [*B.*] *algoritergicola* and [*B.*] *argentinensis* grouped together but not with the type species *B. paragorgiae*. This allowed the authors to propose a new genus, “*Algorimicrobium*” encompassing the eight species, naming the genus after the first described species of this group, [*B*.] *algoritergicola* [1]. However, at the time of writing, the name of the new genus had not yet been approved; therefore, we continue to use its name “*Algorimicrobium*” as a synonym.

In this study, ten genomic sequences of *Bizionia* type strains, including [*B.*] *arctica,* were used for a comprehensive phylogenomic analysis. The list of these strains is given in Table 1. The genome sequence of [*B.*] *psychrotolerans* PB-M7^T^ was not taken in the analysis due to the small quantity of genomic data (GCA_001039595.1, ASM103959v1).

To determine species boundaries, overall genomic relatedness indices (OGRIs) between 041-53-Ur-6^T^ (= KMM 8389ᵀ), *Bizionia,* and “*Algorimicrobium*” type strains were assessed. The ANI values between strain KMM 8389ᵀ and “*A. algoritergicola*” ([*B.*] *algoritergicola*) APA-1^T^ were 80.1% and 78.5–80% with other members of the “*Algorimicrobium*” clade. The strain KMM 8389ᵀ showed the ANI values of 77.4–77.9% with the type species *B. paragorgiae* and other members of the genus *Bizionia*. The dDDH values (formula d4) between strain KMM 8389ᵀ and closely related strains were below 26.7%, which was found for “*A*. *algoritergicola*”. These values were significantly below the recommended ANI and dDDH thresholds for species, which are 95–96% and 70%, respectively. The AAI values between KMM 8389ᵀ and the representatives of the “*Algorimicrobium*” clade ranged from 77.1% (“*A. sediminis*” KCTC 42587^T^) to 81% (“*A. algoritergicola*” APA-1^T^), while with type species *B. paragorgiae* and other members of the genus *Bizionia* they ranged from 69.5% (*B. saleffrena* HFD^T^) to 72.8% ([*B*.] *arctica* CGMCC 1.12751^T^).

To gain an understanding of the phylogenomic relationships within and between the genera *Bizionia* and “*Algorimicrobium*”, genomes of type strains of related genera *Formosa*, *Xanthomarina,* and *Hanstruepera* were additionally selected (Figure 2). The genomic tree based on the 400 conserved proteins [44] clearly exhibited the polyphyletic nature of the genus *Bizionia*, as did the 16S rRNA gene tree in Figure 1. The phylogenomic tree demonstrated 100% branch support for the *Bizionia* and “*Algorimicrobium*” clades as separate genera. Furthermore, [*B.*] *arctica* as well as *Formosa maritima* should be transferred to the genus *Xanthomarina.* The intra-genus AAI values for “*Algorimicrobium*” and *Hanstruepera* were 77.8–88.4 and 80.4–92.2%, respectively, while the inter-genus AAI values between them were 74.0–76.4%, which is slightly lower than the inter-species AAI values observed for each genus.

These results confirm the phylogenetic heterogeneity of *Bizionia* and the clear separation between the genera “*Algorimicrobium*” and *Hanstruepera*. The resulting tree showed that KMM 8389ᵀ forms a distinct species-level lineage (Figure 2).

Thus, the phylogenetic analysis based on the OGRIs and the phylogenetic trees clearly supported KMM 8389^T^ as a novel species of the genus “*Algorimicrobium*” in the family *Flavobacteriaceae*.

### 3.2. Genomic Characteristics and Pan-Genome Analysis

The complete de novo assembled genome of KMM 8389^T^ is a single circular chromosome measuring 3,082,442 bps (Figure 3). The genome G + C content was 33.5%. Annotation through NCBI PGAP [42] identified a total of 2901 genes, comprising 2702 protein-coding sequences, 42 tRNA genes, and 12 rRNA genes arranged in four *rrn* operons. Four 16S rRNA gene sequences retrieved from the genome assembly were 100% identical to those amplified by PCR (PX654852). The chromosomal replication origin, *mnmG*, was located using Ori-Finder 2022 [43] and validated by the GC skew analysis (Figure 3, Appendix A). The KMM 8389^T^ genome sequencing data correspond to the updated minimal standards used in current bacterial taxonomy [55,56].

Metabolic pathway reconstruction of *Bizionia* and “*Algorimicrobium*” members was performed with the Anvi’o pangenomic workflow platform. Analysis revealed a total of 8441 gene clusters comprising 33,288 gene calls (Figure 4a). These gene clusters were categorized into distinct groups as follows: core (1456 gene clusters, 16,256 gene calls), shell (284 gene clusters, 2669 gene calls), cloud (836 gene clusters, 1861 gene calls), *Bizionia* shell (1032 gene clusters, 3300 gene calls), “*Algorimicrobium*” shell (974 gene clusters, 5343 gene calls), and singletons (3859 gene clusters, 3859 gene calls), which potentially included a part of gene clusters from [*B.*] *arctica* shell and cloud. Additionally, reconstruction conducted only for genomes of “*Algorimicrobium*” (Figure 4b) showed a total of 5974 gene clusters comprising 20,841 gene calls with core (1686 gene clusters, 11,978 gene calls), shell (724 gene clusters, 3810 gene calls), cloud (1011 gene clusters, 2500 gene calls), and singletons (2553 gene clusters, 2553 gene calls).

### 3.3. In Silico Analysis of Hydrolytic and Biosynthetic Potentials

The genomic potential for complex carbohydrate degradation was evaluated through mining CAZymes using the dbCAN3 server [49]. The CAZyme proportion in “*Algorimicrobium*” and *Bizionia* genera was relatively low compared to other groups within the family *Flavobacteriaceae* [58] and ranged from 2.25 to 3.33% with the maximum amount predicted in strains KMM 8389^T^, DSM 23925^T^, and CGMCC 1.12751^T^. The most abundant CAZyme family was glycosyltransferase (GT, up to 71), followed by glycoside hydrolase (GH, up to 20) and carbohydrate esterases (CE, up to 5) (Figure 5A). The most common GH among “*Algorimicrobium*” and *Bizionia* type strains belonged to the GH23 family, involved in peptidoglycan degradation. The CAZy family analysis also revealed a shared prevalence of family GH13 exclusively in KMM 8389^T^, DSM 23925^T^, and CGMCC 1.12751^T^, which can digest starch, glycogen, and related marine oligo- and polysaccharides [59]. This finding is confirmed by biochemical tests in which KMM 8389^T^ was able to degrade starch. Moreover, GH31 and GH65 genes also predicted only in these genomes together with GH13 are known to be responsible for targeting maltose in algal polysaccharides.

GTs, especially GT2, GT4, and GT51, were among the most represented families. Notably, they were found in all strains, with GT2 and GT4 reaching copy numbers as high as 30, implying a significant role in the biosynthesis of glycan structures such as exopolysaccharides (EPS), cell wall polymers, or glycoproteins [60].

Compared with the GT and GH, the CE, AA (auxiliary activity), and PL (polysaccharide lyase) families presented limited numbers and diversity, with PLs nearly exclusive to KMM 8389^T^. The presence of five PLs from the families PL 6, 7, 12, and 17 in the genome KMM 8389^T^ indicates specialization toward alginate, corroborating the potential adaptation of this strain to algal substrates and surfaces. Genomes of “*Algorimicrobium*” and *Bizionia* strains encoded a narrow set of CEs from CE3, CE4, CE11, and CE14 families. Most of them presumably have deacetylase activity against N-acetylglucosamine-containing compounds. The strain KMM 8389^T^ lacks the AA genes.

The genomic repertoire of BGCs of the *Bizionia* and “*Algorimicrobium*” genera was evaluated using the antiSMASH server [51]. Seven types of BGCs were identified and are putatively involved in the production of terpenes, type III polyketide synthases (T3PKSs), arylpolyenes, quinone isoprenoid chain, saccharides, fatty acids, and terpene precursors (Figure 5B). Among these BCGs, fatty acid and saccharide BGCs were the most abundant. A BGC linked to flexirubin production, a hallmark of the family *Flavobacteriaceae*, was detected in all genome sequences except for [*B.*] *sediminis* KCTC 42587^T^. Furthermore, BGC, associated with carotenoid biosynthesis, was also identified in all studied genomes, albeit with low similarity confidence. These findings are consistent with the observed yellow-orange pigmentation in all bacterial strains.

The Protologger functional analysis [53] identified 2675 coding sequences in the KMM 8389^T^ strain, including 95 transporters, 16 secretion genes, and 601 unique enzymes; no CRISPR arrays were found. The genome has a pathway for the biosynthesis of folate (vitamin B9) from 7,8-dihydrofolate (EC:1.5.1.3). Also, strain KMM 8389^T^ was predicted to be able to produce zeaxanthin, which is consistent with yellow-colored bacterial colonies. Cbb3-type cytochrome C oxidase was predicted in the genome based on the presence of subunits I, II, III, and IV.

The ecological preferences and distribution of the strain were also examined using the Protologger web tool. No MAGs of the proposed novel species were found when screening thousands of MAGs, using Protologger. Analysis of the 16S rRNA genes showed that the gene sequence of KMM 8389^T^ was represented mostly in coral metagenomes (23.60% of samples), followed by marine sediment (19.50% of samples) and marine (12.50%) metagenomes, which is consistent with its isolation source.

The *Bacteroidota*-specific secretion system IX (T9SS), particularly associated with the gliding motility [61], was detected in “*Algorimicrobium*” and *Bizionia* genomes using the MacSyFinder v. 2 [52]. The studied strains contained genes encoding all the necessary components of the canonical T9SS assembly (*gldJ*, *gldK*, *gldL*, *gldM*, *gldN*, *sprA*, *sprE*, *sprT*, *porQ*, *porU*, and *porV*). Despite this, gliding motility was not observed for the KMM 8389^T^ cells as well as for other “*Algorimicrobium*” species. The strain *B. paragorgiae* DSM 23842^T^ additionally possessed genes of the type VI secretion system (T6SS), which is known as a weapon against Gram-negative bacteria in inter-bacterial competition and in invasion of host cells [62]. Moreover, the strains [*B.*] *argentinensis* JUB59^T^ and [*B.*] *echini* DSM 23925^T^ contained two sets of genes for the secretion system I (T1SS) widely distributed among Gram-negative bacteria and frequently associated with efflux mechanisms [63].

### 3.4. Phenotypic Characterization and Chemotaxonomy

The cells of the KMM 8389^T^ bacterium are small, rod-shaped, 0.7 µm wide and 1–2 µm long, and nonmotile. (Figure 6). Novel bacterium could grow in salinities from 0,5 to 8%. It grew well on/in SWM, MA 2216, and MB 2216.

The novel strain KMM 8389T showed a positive reaction to the hydrolysis of gelatin, DNA, and Tweens 20, 40, 80, as well as type strains studied in this work (Table 2). At the same time, there were differences: strains KMM 8389T, [*B.*] *algotergicola* CIP 108533T, and [*B.*] *echini* KMM 6177T showed a positive reaction to casein hydrolysis and were able to produce H2S, in contrast to [*B.*] *myxarmorum* CIP 108535T. In addition, strains KMM 8389T and [*B.*] *echini* KMM 6177T were able to hydrolyze starch. The strain KMM 8389T was able to grow at 4–35 °C and pH of 6–8.5. Moreover, [*B.*] *algotergicola* CIP 108533T was able to grow at 4–33 °C and pH 6–10. The strain [*B.*] *myxarmorum* CIP 108535T was able to grow at 4–30 °C and pH 6–9. [*B.*] *echini* KMM 6177T was able to grow at 4–36 °C, 0.5–5% NaCl, and pH 6–8.5.

It should be noted that the strains KMM 8389^T^, CIP 108533^T^, CIP 108535^T^, and KMM 6177^T^ studied in this work had differences in phenotypic properties in comparison with type strains of other species, which are representatives of the genus “*Algorimicrobium*”, and the type species of the genus *Bizionia*, *B. paragorgiae* KMM 6029^T^.

The major respiratory quinone is MK-6. Predominant fatty acids of strain KMM 8389^T^ were detected to be iso-C_15:1_ *ω10c* (16.82%), iso-C_15:0_ (15.42%), and iso-C_15:0_ Δ2-OH (11.41%) followed by C_15:0_ (6.04%), iso-C_17:1_ *ω*7*c* (4.49%), iso-C_14:0_ (4.40%), anteiso-C_15:0_ (4.01%), C_15:1_ *ω*6*c* (3.87%), C_16:1_ *ω*7*c* (3.82%), and iso-C_16:1_ *ω*6*c* (3.45%), as shown in Table 3. The other type strains studied in this work showed their own characteristics in the fatty acid profiles. Thus, for [*B*.] *algotergicola* CIP 108533^T^, predominant fatty acids were detected to be C_15:0_ (17.9%), iso-C_15:0_ (12.91%), and iso-C_15:1_ *ω*10*c* (11.26%), followed by iso-C_15:0_ Δ2-OH (6.69%), iso-C_14:0_ (4.97%), iso-C_16:0_ (4.57%), anteiso-C_15:0_ (4.54%), C_15:1_ *ω*11*c* (4.51%), and anteiso-C_15:0_ Δ3-OH (3.26%). For [*B*.] *myxarmorum* CIP 108535^T^, predominant fatty acids were detected to be anteiso-C_15:0_ (26.48%) and iso-C_15:0_ (12.31%), followed by iso-C_16:1_ *ω*6*c* (7.28%), iso-C_15:1_ *ω*10*c* (5.83%), C_15:1_ *ω*6*c* (5.54%), C_15:0_ (5.25%), iso-C_17:1_ *ω*7*c* (4.41%), C_16:1_ *ω*7*c* (4.40%), and iso-C_16:0_ (3.58%). Whereas for the type strain [*B*.] *echini* KMM 6177^T^, predominant fatty acids were detected to be iso-C_15:0_ (21.17%), iso-C_15:0_ Δ2-OH (17.71%), and iso-C_15:1_ *ω*10*c* (15.58%), followed by C_15:0_ (9.23%) and iso-C_17:1_ *ω*7*c* (4.47%) (Table 3).

The major polar lipids of the strain KMM 8389^T^ were a phosphatidylethanolamine (PE), a phosphatidylglycerol (PG), two unidentified lipids (L3, L4), and an unidentified aminolipid (AL1); minor amounts of phosphatidylcholine (PC), an unidentified aminolipid (AL2), an unidentified aminophospholipid (APL), and three unidentified lipids (L1, L2, L5) (Appendix A). The polar lipid profiles of the type strains [*B*.] *algoritergicola* CIP 108533^T^ and [*B.*] *myxarmorum* CIP 108535^T^ differed from that of the new strain KMM 8389^T^ by the presence of an additional unidentified aminolipid (AL2). In the polar lipid profile of the type strain [*B.*] *echini* KMM 6177^T^, the main lipid was also phosphatidylcholine (PC), and in minor quantities, in addition to those described above, two unidentified lipids (L6, L7) were present. At the same time, in the polar lipid profile of the type strain of [*B.*] *myxarmorum* CIP 108535^T^, in addition to those described above, three unidentified lipids (L6, L7, L8) were present in minor quantities.

The DNA G + C content of 33.5% was measured from the genomic sequence of strain KMM 8389^T^, which is close to the values of 33.4–37 mol% calculated for the members of the genus “*Algorimicrobium*”. The phylogenetic uniqueness of the strain KMM 8389^T^ is confirmed by phenotypic differences in temperature and salinity ranges, which determine its growth, ability to hydrolyze substrates, and features of the use of carbohydrates. Differential phenotypic and physiological characteristics are indicated in Table 2 and Appendix A. Based on the combination of phylogenetic analyses and phenotypic characteristics, it is proposed to classify strain KMM 8389^T^ as a novel species, *Algorimicrobium bowmanii*.

## 4. Conclusions

The genomic features of the novel genus and species indicate biotechnological relevance. The presence of polysaccharide-degrading enzymes targeting starch, glycogen, maltose-containing oligosaccharides, and alginate suggests potential for the conversion of algal and other marine polysaccharide substrates. In addition, the abundance of glycosyltransferases points to (exo)polysaccharide production, while the predicted biosynthesis of carotenoids and flexirubin highlights possible applications as sources of natural pigments.

Based on the phylogenomic analysis of all representatives of the genus *Bizionia* and the new strain KMM 8389^T^, as well as on the basis of the conclusions described by Marina Gorcia-Lopez et al. [1], below is a description of the new genus *Algorimicrobium* and its members.

**Description of *Algorimicrobium*, gen. nov.** [1]

Al.go.ri.mi.cro’bi.um (L. masc. n. algor/-oris, cold; N.L. neut. n. *microbium*, a microbe; N.L. neut. n. *Algorimicrobium*, a cold microbe).

Gram-negative, rod-shaped cells, non-motile by gliding. Strictly aerobic and chemoorganotrophic metabolism. Oxidase and catalase are variable. The major menaquinone is MK-6. The major fatty acids include iso-C15:0, and either anteiso-C15:0, iso-C15:1 G or iso-C17:0 3-OH. The genomic G + C content provided in literature is around 33.4–45 mol%. The type species is *Algorimicrobium algoritergicola*, comb. nov.


**Description of *Algorimicrobium algoritergicola*, comb. nov. [1]**


A. al.go.ri.ter.gi’co.la (L. n. *algor*, the cold; L. n. *tergum*, outer covering or surface; L. suff. -*cola*, the dweller, inhabitant; N.L. neut. n. *algoritergicola*, the inhabitant of a cold surface or covering).

Basonym: *Bizionia algoritergicola* Bowman and Nichols 2005

The description is for *Bizionia algoritergicola* [6]. The type strain is APA-1 = ACAM 1056 = CIP 108533 = KMM 8430.


**Description of *Algorimicrobium argentinense*, comb. nov. [1]**


A. ar.gen.ti.nen’se (N.L. neut. adj. *argentinense*, pertaining to Argentina, the country associated with the scientific station in the vicinity of which the strain was isolated).

Basonym: *Bizionia argentinensis* Bercovich et al. 2008

The description is as for *Bizionia argentinensis* [7]. The type strain is JUB59 = CCM-A-29 1259 = DSM 19628.


**Description of *Algorimicrobium echini*, comb. nov. [1]**


A. e.chi’ni (L. gen. n. *echini*, of/from a sea urchin).

Basonym: *Bizionia echini* Nedashkovskaya et al. 2010

The description is as for *Bizionia echini* [8]. The type strain is KMM 6177 = DSM 23925 = KCTC 22015.


**Description of *Algorimicrobium hallyeonense*, comb. nov. [1]**


A. hal.ly.e.o.nen’se (N.L. neut. adj. *hallyeonense*, pertaining to HallyeoMarine National Park, the location of Tongyoung, where the type strain was isolated).

Basonym: *Bizionia hallyeonensis* Yoon et al. 2013

The description is as for *Bizionia hallyeonensis* [9]. The type strain is T-y7 = CCUG 62110 = KCTC 23881.


**Description of *Algorimicrobium myxarmorum*, comb. nov. [1]**


A. myx.ar.mo’rum (Gr. n. *myxa*, slime; L. gen. pl. n. *armorum*, defensive armor; N.L. gen. pl. n. *myxarmorum*, f armor slime, i.e., of the slime on the carapace of a crustacean host).

Basonym: *Bizionia myxarmorum* Bowman and Nichols 2005

The description is for *Bizionia myxarmorum* [6]. The type strain is ADA-4 = ACAM 1058 = CIP 108535 = KMM 8431.


**Description of *Algorimicrobium psychrotolerans*, comb. nov. [1]**


A. psy.chro.to’le.rans (Gr. adj. *psychros*, cold; L. part. adj. *tolerans*, tolerating; N.L. part. adj. *psychrotolerans*, tolerating cold temperature).

Basonym: *Bizionia psychrotolerans* Song et al. 2015

The description is as for *Bizionia psychrotolerans* [10]. The type strain is PB-M7= JCM 19924 = KCCM 43042.


**Description of *Algorimicrobium sediminis*, comb. nov. [1]**


A. se.di’mi.nis (L. gen. n. *sediminis*, of sediment).

Basonym: *Bizionia sediminis* Zhang et al. 2017

The description is as for *Bizionia sediminis* [11]. The type strain is P131 = KCTC 42587 = MCCC 1H00124.


**Description of *Algorimicrobium bowmanii* sp. nov.**


*Algorimicrobium bowmanii* (bow.ma’ni.i. N.L. gen. masc. n. *bowmanii*, of Bowman, in honour of the microbiologist John P. Bowman, who has made important contributions to our knowledge of the diversity of bacteria of the family *Flavobacteriaceae*).

Yellow-pigmented, rod- or ovoid-shaped cells, 0.7 μm in diameter and 1.8–1.4 μm in length, and encapsulated and nonmotile. Growth on/in MA 2216, MB 2216, SWM. Growth occurs in 0.5–8% NaCl (optimal is 0.5–3%), and at 4–35 °C (optimal is 25–28 °C). The pH range for growth is 6.0–8.5 with an optimum of 7.0–8.0. Oxidase and catalase activities are present. Positive for hydrolysis of gelatin, casein, starch, DNA, Tweens 20, 40, 80, and H_2_S production, negative for hydrolysis of the L-tyrosine and nitrate reduction in conventional tests.

According to the API 20E test, it was positive for gelatin hydrolysis and negative for β-galactosidase, arginine dihydrolase, lysine decarboxylase, ornithine decarboxylase, citrate utilization, H_2_S, and urease production under anaerobic conditions, tryptophane deaminase, indole production, acetoin production (Voges–Proskauer reaction), and oxidation of D-glucose, D-mannitol, inositol, D-sorbitol, L-rhamnose, D-sucrose, D-melibiose, amygdalin, and L-arabinose.

Susceptible to (content per disc): ofloxacin (5 µg), rifampicin (5 µg), streptomycin (30 µg), cephalexin (30 µg); ampicillin (10 µg), benzylpenicillin (10 U), vancomycin (30 µg), carbenicillin (100 µg), lincomycin (15 µg), chloramphenicol (30 µg), erythromycin (15 µg); and oleandomycin (15 µg), resistant to: gentamicin (10 µg), neomycin (30 µg), kanamycin (30 µg), nalidixic acid (30 µg), oxacillin (10 µg), cephazolin (30 µg), tetracycline (30 µg), polymyxin B (300 U), doxycycline (10 µg). The DNA GC content of 61.4–61.5% is calculated from the genome sequence. The major respiratory quinone is MK-6. Major fatty acids are iso-C_15:1_ *ω*10*c* (16.82%), iso-C_15:0_ (15.42%), and iso-C_15:0_ Δ2-OH (11.41%). The polar lipids included a phosphatidylethanolamine, a phosphatidylglycerol, five unidentified lipids, two unidentified aminolipids, a phosphatidylcholine, and an unidentified aminophospholipid.

The DDBJ/GenBank accession number for the 16S rRNA gene sequence of strain KMM 8389^T^ is PX654852.

The GenBank accession number for the whole-genome sequence of strain KMM 8389^T^ is JBSTRT000000000.

The type strain is 041-53-Ur-6^T^ = KMM 8389^T^ (=KCTC 72011^T^), isolated from the cavity fluid of the sea urchin *Strongylocentrotus intermedius* collected in the Vries Strait (45.500000, 148.941668), Iturup Island, Kuril Islands, Sea of Okhotsk, Russia.

## Figures and Tables

**Figure 1 microorganisms-14-00024-f001:**
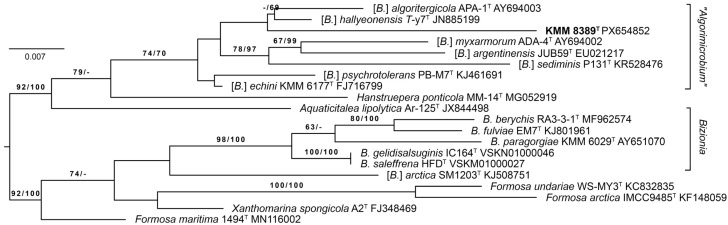
ML/MP 16S rRNA phylogenetic tree showing the position of the novel strain 041-53-Ur-6^T^ (= KMM 8389ᵀ) (in bold) among type strains of the genus *Bizionia*. The ML tree was inferred under the GTR + GAMMA model. The numbers (ML/MP) show bootstrap values greater than 60% measured with 1000 replicates. The bar shows 0.007 substitutions per nucleotide position. GenBank/EMBL/DDB accession numbers are given nearby.

**Figure 2 microorganisms-14-00024-f002:**
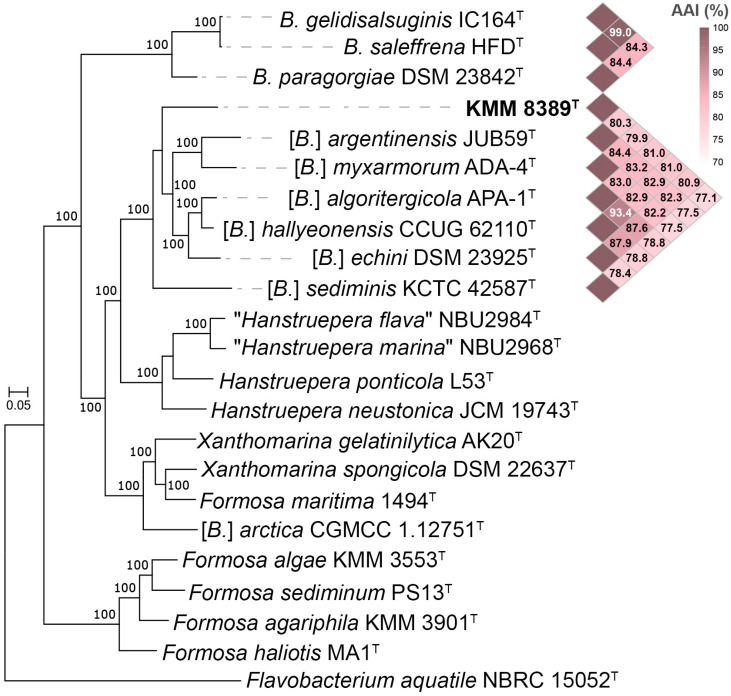
ML genomic tree based on concatenated sequences of 400 conserved proteins showing phylogenetic position of strain KMM 8389^T^ among type strains of *Bizionia*, “*Algorimicrobium*”, and related genera *Formosa*, *Xanthomarina,* and *Hanstruepera*. Bootstrap values are based on 100 replicates. Bar—0.05 substitutions per amino acid position. The AAI values of the *Bizionia* and “*Algorimicrobium*” clades are shown as a heatmap. Strain *Flavobacterium aquatile* NBRC 15052^T^ was used as an outgroup.

**Figure 3 microorganisms-14-00024-f003:**
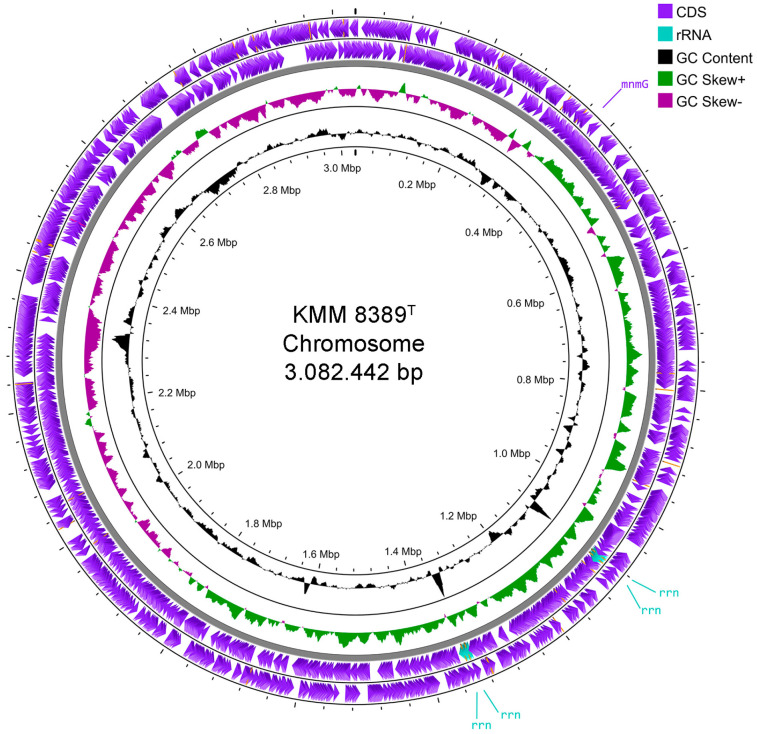
Chromosome map of KMM 8389^T^ created using the Proksee server [57]. The scales are shown on the inside circles in kilobases (Kbp). The figure also shows *rrn* operons (light blue label) and *oriC* (mnmG) (violet).

**Figure 4 microorganisms-14-00024-f004:**
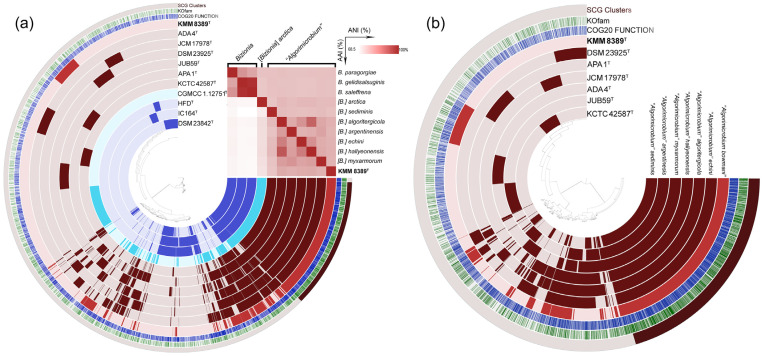
The pan-genome of members of *Bizionia* (**a**) and “*Algorimicrobium*” (**b**) genera generated with anvi’o [48]. Circle bars represent the presence/absence of pan-genomic clusters in each genome. The heatmap in the upper right corner shows pairwise values of ANI and AAI. The strain KMM 8389^T^ is colored in red, other species for “*Algorimicrobium*” in dark red, [*B*.] *arctica* colored in light blue, members of *Bizionia* in blue. Other information included in the figure comprises the COG20 Function and KOfam modules.

**Figure 5 microorganisms-14-00024-f005:**
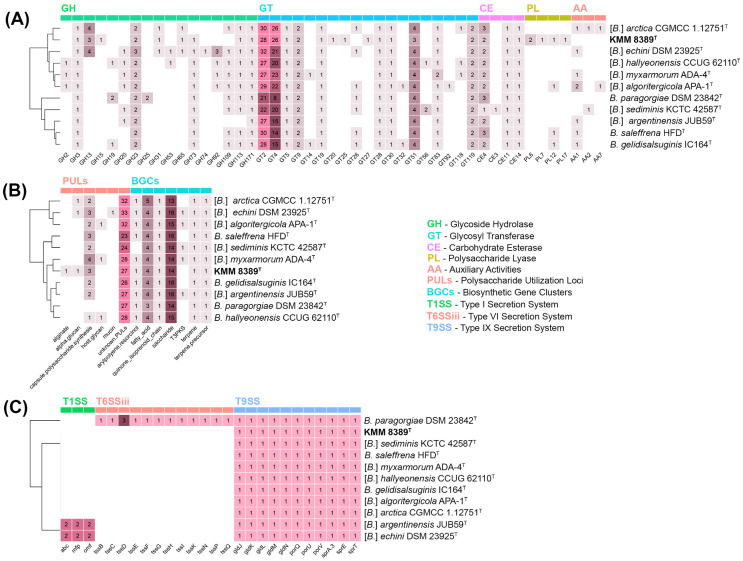
The distribution of CAZymes, PULs, biosynthetic, and secretion system gene clusters in KMM 8389^T^, and type strains of “*Algorimicrobium*” and *Bizionia* genera: (**A**) Heatmap of the CAZyme family abundance. (**B**) Heatmap of the PULs and biosynthetic gene clusters. (**C**) Heatmap of the secretion system gene clusters.

**Figure 6 microorganisms-14-00024-f006:**
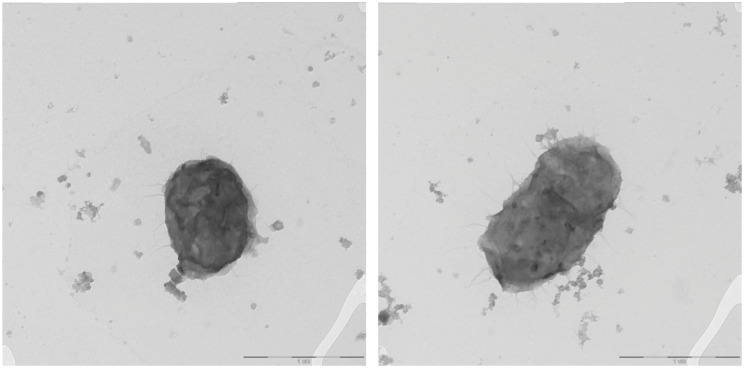
Transmission electron micrographs of strain KMM 8389^T^. Bar—1 µm.

**Table 1 microorganisms-14-00024-t001:** Genomic features of new strain KMM 8389^T^ and type strains of the genera *Bizionia* and *“Algorimicrobium*”.

Feature	1	2	3	4	5	6	7	8	9	10	11
Assembly level	Chromosome	Scaffold	Scaffold	Contig	Contig	Contig	Scaffold	Contig	Contig	Scaffold	Contig
Genome size (Mb)	3.0	3.3	3.3	3.5	3.4	3.3	2.9	3.3	3.4	3.1	3.9
Number of contigs	1	31	30	29	19	70	36	67	118	45	22
G + C Content (mol%)	33.5	33.5	34.5	34	34	34	37	35	35	35.5	33
N50 (Kb)	-	446.7	565.1	679.8	2000	112.3	326.4	217.1	147.8	180.6	1100
L50	1	2	2	2	1	9	2	6	8	8	2
Coverage (x)	315	474	65	219	200	34	158	201	220	436	340
Total genes	2901	3057	3003	3289	3103	3041	2623	2965	3094	2900	3499
Protein-coding genes	2702	2990	2946	3229	3034	2972	2561	2895	3018	2838	3439
rRNAs (5S/16S/23S)	4/4/5	4/3/4	1/1/1	1/1/1	2/2/2	1/1/1	2/1/1	2/1/1	1/1/2	3/1/2	3/0/0
tRNA	42	40	36	36	38	35	36	37	37	35	38
checkM completeness (%)	100.0	99.68	99.68	99.68	99.68	99.02	99.68	100.00	100.00	100.00	99.68
checkM contamination (%)	0.11	0	0.16	0.32	0.16	0.00	0.00	0.00	0.00	0.00	0.32
WGS project	-	FOVN01	JBHSLA01	VSKL01	VSKK01	AFXZ01	JBHULS01	VSKM01	VSKN01	FNQK01	BMFQ01
Genome assembly name	JBSTRT000000000	IMG-taxon 2622736504	ASM4265738v1	ASM808616v1	ASM808620v1	BizArg_ 1.0	ASM4268458v1	ASM808617v1	ASM808618v1	IMG-taxon 2622736593	ASM1463891v1

Strains: **1**, KMM 8389^T^; **2**, [*B.*] *echini* DSM 23925^T^; **3**, [*B.*] *hallyeonensis* CCUG 62110^T^; **4**, [*B.*] *algoritergicola* APA-1^T^; **5**, [*B.*] *myxarmorum* ADA-4^T^; **6**, [*B.*] *argentinensis* JUB59^T^; **7**, [*B.*] *sediminis* KCTC 42587^T^; **8**, *B. saleffrena* HFD^T^; **9**, *B. gelidisalsuginis* IC164^T^; **10**, *B. paragorgiae* DSM 23842^T^; **11**, [*B.*] *arctica* CGMCC 1.12751^T^.

**Table 2 microorganisms-14-00024-t002:** Differential characteristics of strain KMM 8389^T^ and type strains of “*Algorimicrobium*” and *Bizionia*.

Feature	1	2	3	4	5	6	7	8	9
Growth at/in:									
2 °C	ND	ND	ND	ND	+	ND	ND	ND	ND
4 °C	+	+	+	+	+	+	+	+	+
10 °C	+	+	+	+	+	+	+	+	+
28 °C	+	+	+	+	+	+	+	+	+
30 °C	+	+	+	+	−	+	+	+	+
33 °C	+	+	−	+	−	+	−	+	+
35 °C	+	−	−	+	−	+	−	+	+
36 °C	−	−	−	+	−	−	−	+	+
45 °C	−	−	−	−	−	−	−	+	−
0% NaCl	−	−	−	−	−	+	−	+	−
0.5% NaCl	+	+	+	+	−	+	−	+	−
1% NaCl	+	+	+	+	+	+	−	+	+
2% NaCl	+	+	+	+	+	+	+	+	+
3% NaCl	+	+	+	+	+	+	+	+	+
5% NaCl	+	+	+	+	+	+	+	+	+
6% NaCl	+	+	+	−	+	+	+	+	+
8% NaCl	+	+	+	−	−	+	−	+	+
9% NaCl	−	−	−	−	−	+	−	+	−
10% NaCl	−	−	−	−	−	−	−	+	−
pH 5.5	−	−	−	−	ND	+	−	+	
pH 6	+	+	+	+	ND	+	+	+	
pH 8	+	+	+	+	ND	+	+	+	
pH 8.5	+	+	+	+	ND	ND	+	+	
pH 9	−	+	+	−	ND	ND	+	+	
pH 10	−	+	−	−	ND	ND	−	−	
Oxidase	+	+	+	+	+	+	+	−	+
Catalase	+	+	+	−	+	+	−	+	+
Gelatin hydrolysis	+	+	+	+	+	+	+	+	+
Tyrosine hydrolysis	−	−	−	−	+	+	+	ND	ND
DNA hydrolysis	+	+	+	+	−	ND	ND	ND	−
Starch hydrolysis	+	−	−	+	−	−	−	−	−
Casein hydrolysis	+	+	−	+	+	+	+	ND	+
Tweens 20	+	+	+	+	ND	+	+	+	−
Tweens 40	+	+	+	+	ND	+	+	+	+
Tweens 80	+	+	+	+	−	+	+	−	+
H_2_S production	+	+	−	+	−	ND	−	−	+
Nitrate reduction	−	−	−	−	−	−	−	−	−

Strains: **1**, KMM 8389^T^; **2**, [*B*.] *algoritergicola* CIP 108533^T^; **3**, [*B.*] *myxarmorum* CIP 108535^T^; **4**, [*B.*] *echini* KMM 6177^T^. The data were obtained in this work. **5**, [*B.*] *argentinensis* JUB59^T^ [7]; **6**, [*B.*] *hallyeonensis* KCTC 23881^T^ [9]; **7**, [*B.*] *psychrotolerans* KCCM 43042^T^ [10]; **8**, [*B.*] *sediminis* KCTC 42587^T^ [11], **9**, *B. paragorgiae* KMM 6029^T^ [2]. Symbols: (+)—positive, (−)—negative, ND—no data.

**Table 3 microorganisms-14-00024-t003:** Cellular fatty acid composition (% of the total fatty acids) of strains KMM 8389^T^ and type strains of “*Algorimicrobium*” and *Bizionia*.

Fatty Acid	1	2	3	4	5	6	7	8	9
C_14:1_ *ω*10c	0.80	1.03	0.46	0.20					
iso-C_14:0_	4.40	4.79	4.01	1.40	2.0	1.3	1.5		
iso-C_15:1_ *ω*10*c*	16.82	11.26	5.83	15.58	18.1	14.6	10.8		
anteiso-C_15:1_ *ω*11*c*	1.81	1.48	3.18	0.46		1.7	<1		
iso-C_15:0_	15.42	12.91	12.31	21.17	17.3	13.8	22.2	40,7	13.2
anteiso-C_15:0_	4.01	4.54	26.48	1.34	14.0.	2.2	4.8		12.3
C_15:1_ *ω*11*c*	2.92	4.51	0.85	1.77					
C_15:1_ *ω*6*c*	3.87	1.70	5.54	2.08	3.0	1.2	1.1		
C_15:0_	6.04	17.90	5.25	9.23	6.0	5.4			
iso-C_16:1_ *ω*6*c*	3.45	2.42	7.28	1.01		2.8	2.2		5.4
iso-C_16:0_	0.88	4.57	3.58	1.30	1.1	2.0	3.3		5.8
C_16:1_ *ω*7*c*	3.82	2.54	4.40	2.12			11.2		
iso-C_15:0_ Δ2-OH	11.41	6.69	0.06	17.71					
anteiso-C_15:0_ Δ2-OH	1.25	1.21	2.44	0.63					
C_16:0_	0.78	1.34	0.48	1.14		0.9	2.4		
iso-C_15:0_ Δ3-OH	1.33	0.97	0.70	1.32	3.3	3.8	4.5		
anteiso-C_15:0_ Δ3-OH	1.04	3.26	1.09	1.68					
iso-C_17:1_ *ω*7*c*	4.49	1.22	4.41	4.47					
C_17:1_ *ω*8*c*	0.17	1.17	0.55	0.32		0.7	<1		
C_17:1_ *ω*6*c*	0.16	1.13	1.15	0.14		2.8	1.7		
C_16:0_ Δ2-OH	0.75	2.34	1.87	1.33					
iso-C_16:0_ Δ3-OH	2.88	2.84	1.33	1.87	5.2	6.0			6.8
iso-C_17:0_ Δ3-OH	2.16	1.01	0.77	2.11	9.2	11.6	16.7	10,0	

Strains: **1**, KMM 8389^T^; **2**, [*B*.] *algoritergicola* CIP 108533^T^; **3**, [*B.*] *myxarmorum* CIP 108535^T^; **4**, [*B.*] *echini* KMM 6177^T^. The data were obtained in this work. **5**, [*B.*] *argentinensis* JUB59^T^ [7]; **6**, [*B.*] *hallyeonensis* KCTC 23881^T^ [9]; **7**, [*B.*] *psychrotolerans* KCCM 43042^T^ [10]; **8**, [*B.*] *sediminis* KCTC 42587^T^ [11], **9**, *B. paragorgiae* KMM 6029^T^ [2].

## Data Availability

The type strain of the species is strain 041-53-Ur-6^T^ = KMM 8389^T^ (= KCTC 72011^T^). Isolated from the cavity fluid of the sea urchin *Strongylocentrotus intermedius* from the Sea of Okhotsk. The DDBJ/ENA/GenBank accession numbers for the 16S rRNA gene and the whole-genome sequences of strain KMM 8389^T^ are PX654852 and JBSTRT000000000.

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
