# Peer review of "Genome-Based Reclassification of [*Bizionia*] *algoritergicola* Bowman and Nichols 2005 as *Algorimicrobium algoritergicola* gen. nov., comb. nov. and description of *Algorimicrobium bowmanii* sp. nov."

_microorganisms, 2025, doi:10.3390/microorganisms14010024_

Round 1
Reviewer 1 Report
Comments and Suggestions for Authors
This study presents a comprehensive taxonomic investigation that convincingly demonstrates the polyphyly of the genus Bizionia and proposes the creation of a new genus, Algorimicrobium, to accommodate a well-supported clade of eight species, including the newly described Algorimicrobium bowmanii sp. nov. isolated from the sea urchin Strongylocentrotus intermedius. The authors combine state-of-the-art genomic analyses (ANI, AAI, dDDH, phylogenomics), detailed chemotaxonomic profiling (fatty acids, polar lipids, quinones), and thorough phenotypic characterization to support their taxonomic proposals. The work is methodologically sound, clearly written, and fully aligned with current standards in prokaryotic systematics.
Minor Comments
The logical flow of the introduction would be improved by adding a brief, general overview at the beginning. Currently, it starts at a relatively specific point without first establishing the wider scientific context.
The genome has the accession number “XXXXXX”. Please confirm these will be provided upon acceptance.
There is an inconsistency in strain designation: both KMM 8389T and KMM 8393T are used throughout the manuscript (e.g., 8389T in the abstract and methods vs. 8393T in the phylogenetic results). Please verify the correct designation.
The genome of A. bowmanii encodes GH13, GH31, and GH65, which are linked to starch/glycogen/maltose metabolism, and the strain is phenotypically positive for starch hydrolysis. However, it is negative for sucrose, melibiose, and other sugars . Could the authors briefly discuss this specificity?
The discussion could be enhanced by briefly addressing the potential biotechnological relevance of the novel genus and species. Even remarks on possible applications (e.g., enzyme production, bioactive compounds) would be valuable.
The work is thorough, modern, and sets a high standard. I recommend the manuscript for publication after the authors address the minor issues
Author Response
Comment: This study presents a comprehensive taxonomic investigation that convincingly demonstrates the polyphyly of the genus Bizionia and proposes the creation of a new genus, Algorimicrobium, to accommodate a well-supported clade of eight species, including the newly described Algorimicrobium bowmanii sp. nov. isolated from the sea urchin Strongylocentrotus intermedius. The authors combine state-of-the-art genomic analyses (ANI, AAI, dDDH, phylogenomics), detailed chemotaxonomic profiling (fatty acids, polar lipids, quinones), and thorough phenotypic characterization to support their taxonomic proposals. The work is methodologically sound, clearly written, and fully aligned with current standards in prokaryotic systematics.
Response: Thank you very much for taking the time to review our manuscript and for the high appreciation of our work.
Comment 1: The logical flow of the introduction would be improved by adding a brief, general overview at the beginning. Currently, it starts at a relatively specific point without first establishing the wider scientific context.
Response 1: Thanks for the important suggestion. We have added a rationale for this work to the Introduction section.
Comment 2: The genome has the accession number “XXXXXX”. Please confirm these will be provided upon acceptance.
Response 2: The delay in receiving the accession number was caused by a two-month shutdown affecting NCBI. After a personal request, we finally received the genome accession number for this strain, which has been added to the text where necessary.
Comment 3: There is an inconsistency in strain designation: both KMM 8389T and KMM 8393T are used throughout the manuscript (e.g., 8389T in the abstract and methods vs. 8393T in the phylogenetic results). Please verify the correct designation.
Response 3. Thank you very much. We are very sorry for typo. This has been fixed throughout the text and figures.
Comment 4: The genome of A. bowmanii encodes GH13, GH31, and GH65, which are linked to starch/glycogen/maltose metabolism, and the strain is phenotypically positive for starch hydrolysis. However, it is negative for sucrose, melibiose, and other sugars. Could the authors briefly discuss this specificity?
Response 4: The three GH13, GH31 and GH65 are responsible for utilization of α-glucans (e.g. starch, glycogen) as well as for degradation their products such as maltose. This is consistent with the observed positive starch hydrolysis phenotype. We suppose that the absence of specific glycoside hydrolases (e.g. GH32, GH27/GH36) and corresponding transport systems required for their uptake and intracellular catabolism reflects the inability to utilize sucrose, melibiose and related sugars. This fact indicates a substrate preference for polymeric α-glucans rather than low-molecular-weight disaccharides.
Comment 5: The discussion could be enhanced by briefly addressing the potential biotechnological relevance of the novel genus and species. Even remarks on possible applications (e.g., enzyme production, bioactive compounds) would be valuable.
Response 5: Thanks for your suggestion. We have added some points to Conclusion. Please look at the Lines 560-565.
Comment 6: The work is thorough, modern, and sets a high standard. I recommend the manuscript for publication after the authors address the minor issues.
Response 6: Thank you very much for the high appreciation of our manuscript.

Reviewer 2 Report
Comments and Suggestions for Authors
Dear authors, the article contains very interesting microbiological information regarding the taxonomy of Bacteroidota bacteria. My comments are presented in the attached file. The following are key: the authors should structure the Results section according to the Materials and Methods section. A description of the genomic studies should be follow the description of the morphology and biochemistry. The deposition numbers of the nucleotide sequences and genome should be included.

Author Response
Comment 1: Dear authors, the article contains very interesting microbiological information regarding the taxonomy of Bacteroidota bacteria. My comments are presented in the attached file.
Response 1: Thank you very much for taking the time to review our manuscript and for positive assessment of our work.
Comment 2: The following are key: the authors should structure the Results section according to the Materials and Methods section. A description of the genomic studies should be following the description of the morphology and biochemistry.
Response 2: Unfortunately, we do not agree with your suggestion to present the results in the same order as in the Methods section, and vice versa.
When writing this article, we were guided by the following logic: the Methods section begins with the isolation of the strain, followed by a description of the phenotypic methods, so as not to interrupt the microbiology part. In the section results, we started with identification of a microorganism based on 16S rRNA and genome data, in order to determine its nearest neighbors, which are necessary for further biochemical and chemotaxonomic tests.
Comment 3: The deposition numbers of the nucleotide sequences and genome should be included.
Response 3: The delay in receiving the accession number was caused by a two-month shutdown affecting NCBI. After a personal request, we finally received the genome accession number for this strain, which has been added to the text where necessary.
From the attached pdf file:
Comment 4: I would advise authors to indicate the phylum to which the bacterium belongs in the title of the article and in the abstract (Line 19 from microorganisms-4042490-review.pdf).
Response 4: Thank you for the suggestion. We have added the phylum name to the abstract but did not include it in the article title to avoid making the title overly long and heavy.
Comment 5: What will be the type strain of the genus? The genus will have only one type strain (Line 52, microorganisms-4042490-review.pdf).
Response 5: Each bacterial genus has a single type species. For the genus Bizionia, the type species is Bizionia paragorgiae. Each species has one type strain, details of which are provided in the Introduction section below. Lines 51-53 contain information in accordance with the article we are referring to (García-López M. et al. Analysis of 1,000 Type-Strain Genomes Improves Taxonomic Classification of Bacteroidetes. Front Microbiol. 2019; 10: 2083).
Comment 6: Mistake - 16S rRNA (Line 142, microorganisms-4042490-review.pdf)
Response 6: Sorry for the typo, we've corrected it.
Comment 7: I would like to point out to the authors that the sections should be presented in accordance with the description of materials and methods. 3.1 Phenotypic and biochemical description 3.2 Analysis 16, since there will be a larger number of relatives here than in the genomic data 3.3 Genomic study. The current structure is disordered (Line 197, microorganisms-4042490-review.pdf).
Response 7: Look at the response 1, please.
Comment 8: Accession numbers are absent (Line 560, microorganisms-4042490-review.pdf).
Response 8: The lack of accession numbers was due to a delay in receiving them from NCBI due to their temporary shutdown; by now they have been received and included in the manuscript - in the text, table and figures.

Reviewer 3 Report
Comments and Suggestions for Authors
The manuscript of Kurilenko et al. investigates the genome-based reclassification of members of the genus Bizionia and proposes a novel genus, Algorimicrobium, together with the description of a new species, Algorimicrobium bowmanii. Using a comprehensive polyphasic approach that combines phylogenomic, chemotaxonomic, phenotypic, and functional genomic analyses, the authors provide convincing evidence for the phylogenetic heterogeneity of Bizionia and the taxonomic validity of the proposed reclassification. The article is very well written, clearly structured, and the applied methodology is generally sound. The study represents a valuable contribution to microbial systematics and merits publication in Microorganisms. However, after careful evaluation, it is the reviewer’s opinion that the manuscript lies on the border between minor and major revision, as several aspects still require further clarification or elaboration, particularly with regard to specific methodological details that are addressed in my comments below.
- Line 98: briefly mention the procedure of the methods instead of solely referring to the literature.
- Line 109-111 and Line 112-113: please provide a reference.
- Lines 122-123: provide some additional information on the extraction procedure (briefly).
- Lines 126-129: is this a common procedure, if yes, please add a reference.
- Why did the authors apply two different GC methods for the determination of FAMEs? Please elaborate.
- Mind significant figures in the text and tables. Sometimes different significant figures are used for the same parameter. For instance, in Table 1, but not limited to this example. Please go through the text and adapt.
- There is a highlighted text in yellow present in Line 285, the reviewer assumes that the number still needs to be added.
- Lines 327-329: please provide a reference.
- Sometimes a space is used between a number and unit, sometimes not (even for the same parameter, for instance °C). Please go through the text and adapt for consistency.
- Table 2: there are lines under ° in °C. Please remove them.
- Table 3: the authors depict “fatty acid” composition. Did the authors determine the fatty acids via GC or FAMEs? These are two different things. Please specify. Also the results are depicted in %. Is this based on dry weight or on total lipids or on something else? Please specify.
- Line 556: Here, too, there is still highlighted text in yellow.
- Please give some introductory lines in the conclusion section. Currently it reads as a sum-up, also in Lines 496-527.
- Lines 523-524: the GenBank accession number is probably not yet available, since it is indicated with XXXXXX? Please add the number or remove these Lines.
Author Response
Comment 1: The manuscript of Kurilenko et al. investigates the genome-based reclassification of members of the genus Bizionia and proposes a novel genus, Algorimicrobium, together with the description of a new species, Algorimicrobium bowmanii. Using a comprehensive polyphasic approach that combines phylogenomic, chemotaxonomic, phenotypic, and functional genomic analyses, the authors provide convincing evidence for the phylogenetic heterogeneity of Bizionia and the taxonomic validity of the proposed reclassification. The article is very well written, clearly structured, and the applied methodology is generally sound. The study represents a valuable contribution to microbial systematics and merits publication in Microorganisms. However, after careful evaluation, it is the reviewer’s opinion that the manuscript lies on the border between minor and major revision, as several aspects still require further clarification or elaboration, particularly with regard to specific methodological details that are addressed in my comments below.
Response 1: Thank you very much for taking the time to review our manuscript and for the high appreciation of our work.
Comment 2: Line 98: briefly mention the procedure of the methods instead of solely referring to the literature.
Response 2: A brief description of the methods has been added. Please, look at lines 100-103.
Comment 3: Line 109-111 and Line 112-113: please provide a reference.
Response 3: The references under numbers 14, 15 were added.
Comment 4: Lines 122-123: provide some additional information on the extraction procedure (briefly).
Response 4. Additional information was added. Please, look at lines 127-129.
Comment 5: Lines 126-129: is this a common procedure, if yes, please add a reference.
Response 5: The references under numbers 18, 19 were added.
Comment 6: Why did the authors apply two different GC methods for the determination of FAMEs? Please elaborate.
Response 6: GC-FID was used for quantification, GC-MS – for identification of FAs, clarification added.
Comment 7: Mind significant figures in the text and tables. Sometimes different significant figures are used for the same parameter. For instance, in Table 1, but not limited to this example. Please go through the text and adapt.
Response 7: Thank you for your comment. We checked and fixed it throughout the text.
Comment 8: There is a highlighted text in yellow present in Line 285, the reviewer assumes that the number still needs to be added.
Response 8: Thank you for your comment. We finally received the genome accession number for this strain, which has been added to the text where necessary.
Comment 9: Lines 327-329: please provide a reference.
Response 9: The reference under a number 60 was added.
Comment 10: Sometimes a space is used between a number and unit, sometimes not (even for the same parameter, for instance °C). Please go through the text and adapt for consistency. Table 2: there are lines under ° in °C. Please remove them.
Response 10: Thank you for your comment. We checked and fixed it throughout the text.
Comment 11: Table 3: the authors depict “fatty acid” composition. Did the authors determine the fatty acids via GC or FAMEs? These are two different things. Please specify. Also the results are depicted in %. Is this based on dry weight or on total lipids or on something else? Please specify.
Response 11: The fatty acids were analyzed by gas chromatography (GC) method in the form of methyl esters (MEFA) derivatives. The results for the fatty acid content are presented as a % of the total fatty acids obtained by hydrolysis and methylation directly from the containing bacteria medium (MIDI method), clarification added.
Comment 12: Line 556: Here, too, there is still highlighted text in yellow.
Response 12: This has been fixed.
Comment 13: Please give some introductory lines in the conclusion section. Currently it reads as a sum-up, also in Lines 496-527.
Response 13: Thanks for your suggestion. We have added some points to Conclusion. This is a taxonomic article, where the conclusion corresponds to the detailed protologue (a set of associated elements representing the first publication of a new taxon). This type of conclusion is a mandatory part and the basis for validating the publication of the taxon name.
Comment 14: Lines 523-524: the GenBank accession number is probably not yet available, since it is indicated with XXXXXX? Please add the number or remove these Lines.
Response 14: The GenBank accession numbers for this strain have been added to the text.

Round 2
Reviewer 3 Report
Comments and Suggestions for Authors
The authors addressed most of the reviewer's comments, and therefore, the reviewer accepts this manuscript for publication in Microorganisms.